evolution, developmental biology, biomechanics

Serpentes, Ophidia, Colubroidea, fangs, venom groove, development

**Author for correspondence:**
Alessandro Palci
e-mail: alessandro.palci@flinders.edu.au

# Plicidentine and the repeated origins of snake venom fangs

Alessandro Palci[1,2], Aaron R. H. LeBlanc[3,5], Olga Panagiotopoulou[6], Silke G. C. Cleuren[7], Hyab Mehari Abraha[6], Mark N. Hutchinson[1,2], Alistair R. Evans[7,8], Michael W. Caldwell[3,4] and Michael S. Y. Lee[1,2]

[1]College of Science and Engineering, Flinders University, Adelaide, SA 5042, Australia
[2]South Australian Museum, North Terrace, Adelaide, SA 5000, Australia
[3]Department of Biological Sciences, and [4]Department of Earth and Atmospheric Sciences, University of Alberta, Edmonton, AB Canada, T6G 2E9
[5]Faculty of Dentistry, Oral and Craniofacial Sciences, Guy's Campus, King's College London, London WC2R 2LS, UK
[6]Department of Anatomy and Developmental Biology, Monash Biomedicine Discovery Institute, Faculty of Medicine Nursing and Health Sciences, Monash University, Melbourne, VIC 3800, Australia
[7]School of Biological Sciences, Monash University, VIC 3800, Australia
[8]Geosciences, Museums Victoria, Melbourne, VIC 3001, Australia

AP, 0000-0002-9312-0559; ARHL, 0000-0002-2497-1296; OP, 0000-0002-6457-448X; ARE, 0000-0002-4078-4693; MWC, 0000-0002-2377-3925

Snake fangs are an iconic exemplar of a complex adaptation, but despite striking developmental and morphological similarities, they probably evolved independently in several lineages of venomous snakes. How snakes could, uniquely among vertebrates, repeatedly evolve their complex venom delivery apparatus is an intriguing question. Here we shed light on the repeated evolution of snake venom fangs using histology, high-resolution computed tomography (microCT) and biomechanical modelling. Our examination of venomous and non-venomous species reveals that most snakes have dentine infoldings at the bases of their teeth, known as plicidentine, and that in venomous species, one of these infoldings was repurposed to form a longitudinal groove for venom delivery. Like plicidentine, venom grooves originate from infoldings of the developing dental epithelium prior to the formation of the tooth hard tissues. Derivation of the venom groove from a large plicidentine fold that develops early in tooth ontogeny reveals how snake venom fangs could originate repeatedly through the co-option of a pre-existing dental feature even without close association to a venom duct. We also show that, contrary to previous assumptions, dentine infoldings do not improve compression or bending resistance of snake teeth during biting; plicidentine may instead have a role in tooth attachment.

## 1. Introduction

Snakes are unparalleled among vertebrates in their evolution of sophisticated venom delivery systems, which consist of venom glands, highly modified teeth for venom injection and associated muscles and bones [1–3]. Despite their complexity, venom delivery systems evolved in snakes multiple times, resulting in over 600 medically important species [4,5]. There are two main types of snake venom fangs: grooved fangs, where venom spreads down an open anterolateral or lateral groove, and tubular fangs, where venom runs through a canal (or duct) that arises developmentally from the closure of a groove [4]. Based on recent phylogenetic studies, tubular fangs probably evolved independently in three front-fanged clades: Viperidae, Elapidae and Atractaspidinae (a subfamily of Lamprophiidae) [3,6,7]. Despite their convergent origins, the developmental and structural resemblances of these fangs are striking [4,8–10].

Previous developmental studies have revealed that the venom groove or canal forms from precise infoldings of the epithelial wall of the developing tooth germ [11]. These infoldings resemble the development of plicidentine (folded dentine), a term used to describe a series of invaginations at the base of a tooth in several other vertebrate groups [12,13]. These invaginations also originate from infoldings of the epithelium of a developing tooth base, and when dentine is formed by the odontoblasts lining this epithelium it acquires a 'folded' appearance [12,13]. Some form of plicidentine has been reported in fishes (some sarcopterygians and actinopterygians), early tetrapods (e.g. 'labyrinthodont' amphibians), early stem amniotes, early synapsids, ichthyosaurs, choristoderes and 'varanoid' lizards [12–20]. While plicidentine has also been reported in some mosasaurs (extinct marine lizards) and two fossil snakes, these reports have been contested [12,16,21,22]. 'Invaginations' in the dentine at the base of cobra fangs (*Naja naja*) have been previously noted [23], but have not been identified as plicidentine. Some authors [16] identified small dentine infoldings in some basal snakes (henophidians), but argued against their interpretation as plicidentine. So far, the only undisputed cases of plicidentine among living reptiles are in 'varanoid' lizards (i.e. *Varanus*, *Lanthanotus* and *Heloderma* [16]); however, to date, no comprehensive investigation (histological or microCT) of the presence of plicidentine in a broad sample of snakes has been undertaken.

We investigated tooth microstructure across the major snake clades using multiple approaches, in order to (i) verify whether plicidentine is present in snakes; (ii) determine if plicidentine is developmentally correlated with the origin of venom grooves and (iii) formulate a novel hypothesis of snake fang origin and evolution based on the examination of tooth microanatomy (microCT), development (histology and gene expression), and biomechanics (finite-element analysis, FEA).

## 2. Material and methods

### (a) High-resolution computed tomography
We examined high-resolution computed tomography (microCT) scans from 19 snake species and 3 lizard species (all adult specimens; electronic supplementary material, tables S1 and S2), and histological sections from four of these species (three snakes and one lizard; see below under 'Stained histology sections'). Sixteen specimens were microCT scanned specifically for this study, while data from eight other species were available from pre-existing scans at the South Australian Museum (Adelaide, South Australia), including a scan of the basal fossil taxon *Yurlunggur* (electronic supplementary material, tables S1 and S2). Three-dimensional (3D) visualization, imaging and segmentation were carried out using Avizo Lite (v. 7.0 and 9.0; Thermo Scientific) and Dragonfly v. 4.1 (Object Research Systems Inc.).

### (b) Finite-element analysis
For the biomechanical testing of the fang, we selected *Fordonia leucobalia*, the crab-eating water snake [24]. We chose this species because its fangs possess a groove and also large-scale plicidentine infoldings, which are easy to model without introducing modelling errors. We chose *F. leucobalia* over a front-fanged snake because according to some studies, rear fangs could be the evolutionary precursors of front fangs (e.g. [25]), and we wanted to test whether a simple groove (as opposed to a more derived and complex tubular fang) could have initially appeared because it provided a biomechanical advantage during biting. *Fordonia leucobalia* was also chosen because of its durophagous diet [24], which is reflected by its robust dentition, providing a more extreme test of the biomechanical function of the groove and/or plicidentine.

The 3D geometries of the fang of *F. leucobalia* (SAMA R26990) for the biomechanical testing of the groove and plicidentine were extracted in Mimics v. 23.0 software (Materialise, Leuven, Belgium) using a combination of automatic (thresholding) and manual segmentation of CT data acquired at a resolution of 4.034 µm (electronic supplementary material, table S1). Only the portion of the maxilla immediately dorsal to the fang was retained, and the rest was digitally removed producing a vertical cutting plane. To test the mechanical function of the plicidentine and venom groove during biting, we created a series of models where plicidentine infoldings and/or the venom groove were removed while keeping the total dentine volume approximately constant (electronic supplementary material, table S3). This was achieved by editing the 3D mesh file of the original *F. leucobalia* model using automated smoothing and mesh editing functions in 3-Matic v. 15.0 (Materialise, Leuven, Belgium). We created four model variants: (i) the original fang; (ii) the 'no plicidentine' model with only the plicidentine removed; (iii) the 'no groove' model with only the venom groove removed; and (iv) the 'no groove-no plicidentine' model with both the venom groove and the plicidentine removed. All 3D surface models were then converted into volumetric mesh files of solid continuum linear tetrahedral elements of 0.025–0.07 mm size (C3D4) and exported to Abaqus CAE Simulia 2019 (Dassault Systémes, Velizy-Villacoublay) for FEA. The fang was modelled as a solid structure and assigned linear elastic, homogeneous and isotropic material properties from the literature ($E = 20$ GPa and $v = 0.3$ [26,27]).

Due to the lack of published bite force measurement or muscle force data for *F. leucobalia*, we estimated maximum bite force (F) using head height (HH) from a recent study [28] as a proxy of maximum bite force in snakes (regression equation: $\log(F) = 1.12 \times \log(HH) + 0.47$). This produced an estimated bite force of 5.3 N for our original model of *F. leucobalia*. To assign a maximum bite force to the model variants, we scaled the original estimated bite force of 5.3 N to the volume (when evaluating strain distributions) or surface area (when evaluating stress distributions) of each model as per [29] (electronic supplementary material, table S3).

Each fang was constrained (all translations and rotations) anteriorly on the maxilla (i.e. on the vertical cutting plane) and tested under the same four loading cases (loads 1–4). Load 1 involved the application of a compressive force at the tip of the fang along its long axis (+x axis). Load 2 was similarly applied to the tip, but directed perpendicular to load 1 (along the +z axis). This load was applied in order to test the tooth's resistance to lateral bending. We also tested two additional loading cases where the surface area of the load application was expanded to a large portion of the crown and directed vertically to simulate penetration into a prey item (load 3), or anteriorly to simulate a prey item trying to escape (load 4) (electronic supplementary material, figure S1).

All finite-element models were solved using the Abaqus implicit direct static solver and Newtonian default iterations. To evaluate the biomechanical importance of the groove and plicidentine (i.e. whether they increase resistance to compression and bending), we compared Von Mises stress and principal strain regimes between the grooved and non-grooved models and models with and without plicidentine.

### (c) Stained histology sections
Stained histology sections were prepared for the snakes *Boa constrictor*, *Hydrophis cyanocinctus* and *Oxyuranus scutellatus*

(electronic supplementary material, table S2). Sections from *B. constrictor* were made at the Advanced Microscopy Facility in the Department of Biological Sciences, University of Alberta, while the other two snakes were sectioned and stained at Histology Services, Department of Health Sciences University of Adelaide. The maxilla of the *B. constrictor* was dissected from the head of a recently deceased specimen and decalcified in Richard-Allen Scientific CalRite solution (formic acid and formaldehyde) for three weeks. The decalcified specimen was then placed in a dehydration series of toluene and ethanol overnight, and embedded in paraffin wax. The specimen was then sectioned horizontally at 5–8 µm thickness using a microtome and stained using haematoxylin and eosin. The dissected heads of a sea snake (*H. cyanocinctus*) and a coastal taipan (*O. scutellatus*) were fixed in 10% neutral-buffered formalin (NBF; 4.0% formaldehyde in phosphate-buffered saline solution) for 2 days, rinsed in a bath of clean water for 1 day and then placed in 70% ethanol. The heads were then decalcified in 10% EDTA (ethylenediaminetetraacetic acid; made up from powder) for a week, changing the solution every (working) day. The decalcified heads were embedded in paraffin wax and then sliced coronally using a microtome and the slides were stained with haematoxylin and eosin or Gomori's trichrome. Staining protocols follow the guidelines published in previous studies [30,31].

The thin sections from the *B. constrictor* were imaged using a Nikon DS-Fi3 camera mounted to a Nikon Eclipse E600 polarizing microscope and NIS-Elements imaging software. High-resolution images of the histology sections of *H. cyanocinctus* and *O. scutellatus* were taken with a NanoZoomer 2.0HT digital slide scanner (Hamamatsu Photonics) at the Faculty of Health and Medical Sciences of the University of Adelaide and visualized in NDP view v. 2 (Hamamatsu Photonics).

## 3. Results

We have found evidence of plicidentine in maxillary, dentary and palatal teeth of alethinophidians (i.e. all living snakes except blind snakes) (figure 1; electronic supplementary material, figures S2 and S3). Observations of plicidentine in microCT images were confirmed through histology sections to exclude the possibility that the tissue forming the infoldings could be something other than dentine (figure 2; electronic supplementary material, figures S4–S8).

Plicidentine in snakes typically comes in the form of very tight, small infoldings in basal alethinophidians such as *Anilius*, *Cylindrophis*, *Boa* (figure 2a,b; electronic supplementary material, figure S6a,b), *Morelia* and *Liasis* or in fangless colubroids (e.g. *Nerodia rhombifera*) (figure 1; electronic supplementary material, figures S2 and S3). In colubroid snakes, plicidentine is also present in marginal and palatal teeth (e.g. in the rear-fanged *Fordonia leucobalia*), but in front-fanged species it is generally weakly expressed outside of the fangs, with the exception of the largest anterior dentary teeth in some species (e.g. *Bitis gabonica*, *Hydrophis cyanocinctus* and *Oxyuranus scutellatus*; electronic supplementary material, figure S3w). Plicidentine was also present in a fossil basal snake, the madtsoiid *Yurlunggur* (electronic supplementary material, figures S2 and S3f), confirming previous reports [21,22]. Among sampled snake species, plicidentine was absent only in the blind snake *Anilios* (*Ramphotyphlops*) *bicolor* (electronic supplementary material, figure S3 g).

Notably, the largest plicidentine folds are present in the fangs of venomous snakes (figure 1; electronic supplementary material, figures S2, S3 and S6c–d), both front-fanged (viperids, elapids, atractaspidines) and rear-fanged (e.g.

homalopsids). Plicidentine in fangless snakes is always restricted to the tooth base, where the tooth is anchored to the surrounding tissue. However, in venom fangs it extends some distance towards the crown tip and is also expressed externally on the posterior tooth surface as narrow longitudinal crenulations (electronic supplementary material, figure S9) (not to be confused with tooth ornamentation often occurring in piscivorous snakes, which only affects the external surface of the dentine and the enamel layer [32]). The venom groove blends in with the folded pattern of the dentine wall at the base of the tooth, suggesting that the groove is simply a larger and deeper dentine fold, such that the enamel on the external surface of the tooth is also infolded. The histology sections revealed that in developing fangs, the primordium of the venom groove is associated with epithelial crenulations that ultimately demarcate the dentine wall, and the groove itself appears to be simply a much larger infolding (figure 2; electronic supplementary material, figures S5–S7).

During the formation of a reptilian tooth, the developing tooth bud is surrounded by several layers of epithelium (electronic supplementary material, figure S7), which help determine the shape of the tooth prior to any dentine or enamel mineralization. In the crowns of reptile teeth, there are three epithelial layers: the outer enamel epithelium, an intervening layer of widely spaced cells called the stellate reticulum and the inner enamel epithelium. Of these, only the inner enamel epithelium contributes to the formation of the hard tissues of the tooth [33].

In the developing venom fang, the inner enamel epithelium produces the initial shape of the venom groove. This same layer also produces the apices of the largest plicidentine folds, but for the most part, plicidentine is restricted to the base of the teeth. Considering that reptile teeth form from the tip towards the base, and because most plicidentine is restricted to the more basal regions of a tooth, the basal folds do not form before the venom groove, which extends to the tip of the tooth. Basal folds that are restricted to the very base of a tooth are formed by a continuation of the epithelial tissues in the root, Hertwig's epithelial root sheath (HERS) [33] (electronic supplementary material, figures S7 and S8).

We also observed that the development of grooved teeth in the upper jaws of venomous snakes is initially correlated only with the degree of development of plicidentine, and not with the proximity of a venom gland, despite the fact that venom fangs and venom glands develop from the same epithelial primordium (the posterior maxillary dental lamina [25]). In fact, grooved teeth can develop anywhere in the maxilla (see below), and to a lesser degree even in the lower jaw, despite the lack of a connection to a venom gland. The dentary teeth of the examined elapids have prominent infoldings in early stages of tooth development (figure 2f,g), which are associated with a thin groove, or 'furrow' [34], situated on the anterior surface of the crowns in mature teeth (figure 2h; electronic supplementary material, figure S3x). Individual variants of *Oxyuranus scutellatus* (coastal taipans)—where a venom groove appears not only on each fang, but also on the posterior maxillary teeth—further reinforce the developmental connection between plicidentine and the venom groove. Delocalized formation of such grooves is an infrequent yet recurring condition in fanged colubroids [34,35], and we additionally documented it in some other elapid specimens: *Acanthophis antarcticus*, *Bungarus fasciatus*, *Notechis scutatus* and *Oxyuranus*

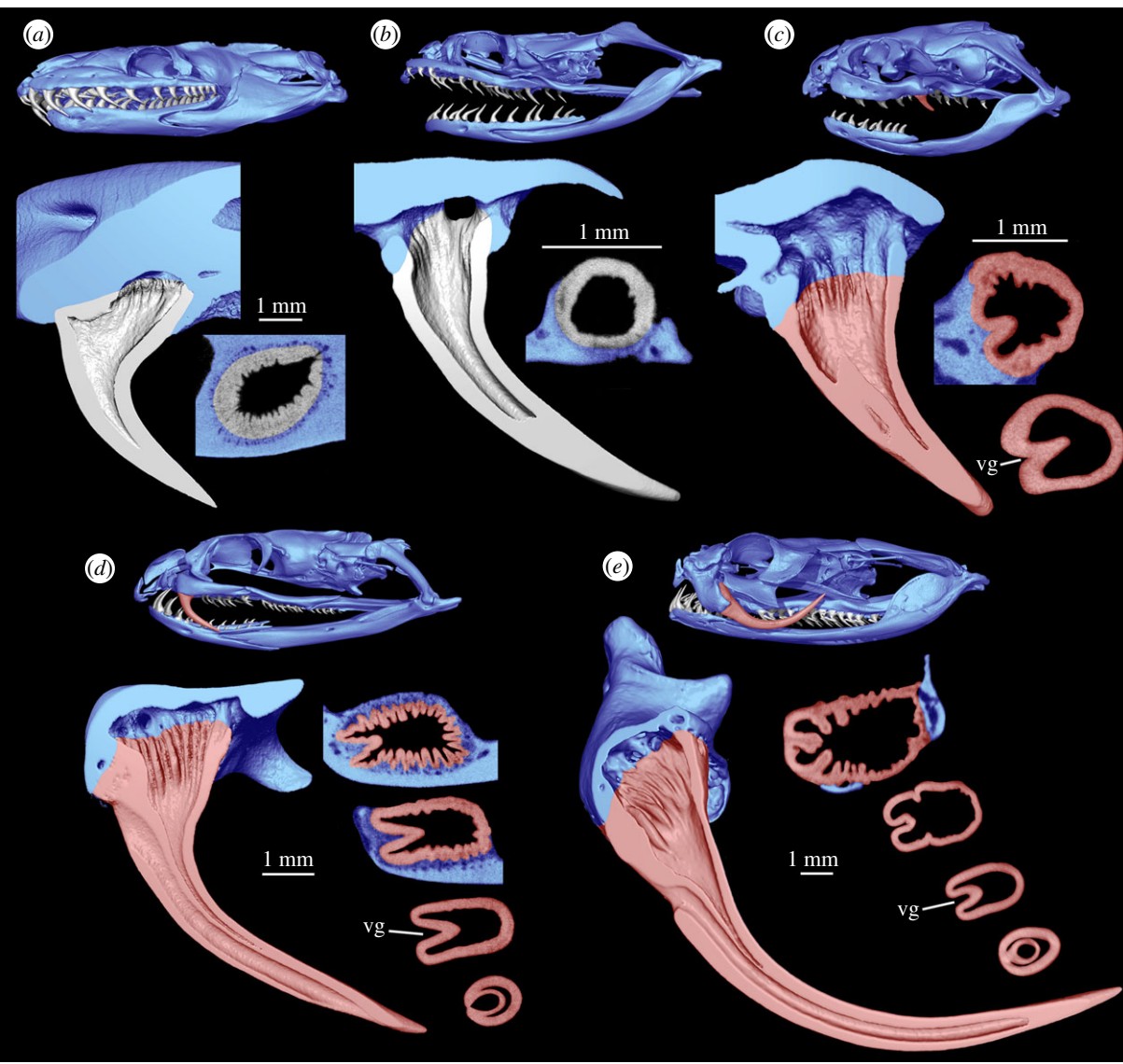

**Figure 1.** Plicidentine is found in most snakes and is particularly evident in venom fangs. Longitudinal and horizontal sections through a selection of snake teeth. (*a*) The pythonid *Liasis olivaceus*. (*b*) The fangless colubrid *Nerodia rhombifera*. (*c*) The rear-fanged homalopsid *Fordonia leucobalia*. (*d*) The front-fanged elapid *Oxyuranus scutellatus*. (*e*) The front-fanged viperid *Bitis gabonica*. Venom fangs are highlighted red, while maxillary and attachment tissues are highlighted blue. The labial side of the tooth and maxilla is removed in the longitudinal sections to expose the pulp cavity and the plicidentine at the base of the tooth. Horizontal sections at the base of the tooth are accompanied by additional sections towards the crown tip in (*c*–*e*). Note how in the horizontal sections the venom groove (vg) is continuous dorsally with a large fold of plicidentine, which in (*d*) and (*e*) is itself affected by second-order infolding within the root.

*microlepidotus* (electronic supplementary material, figure S10 and table S2). We have found that in *O. scutellatus*, these anomalous posterior teeth are associated with unusually extensive development of plicidentine, where the folds are deeper and more numerous than in the other teeth (figure 3). Whereas in normal *O. scutellatus* plicidentine is weakly expressed in the smaller posterior maxillary teeth, in the variant specimen, those teeth show extensive infoldings that parallel those in the fangs. Furthermore, one of these infoldings is much larger than the others and is associated with an enclosed groove that is morphologically identical to the venom groove of the front fang, making these teeth tubular (figure 3*f*,*g*). These grooved teeth in *O. scutellatus* are not connected to terminal venom gland ducts, suggesting that they are developmentally linked to the strong expression of plicidentine rather than to the venom gland.

Plicidentine in snakes resembles the condition in the venomous lizard *Heloderma* (electronic supplementary material, figures S2 and S3), but does not resemble the elaborate honeycomb-like lattice found in the lizard *Varanus* (electronic supplementary material, figures S2, S3 and S8). Previous comparisons between snake and *Varanus* teeth partly explain why some authors argued against the presence of plicidentine in snakes [16]. Interestingly, similarly to snakes, the venom groove in *Heloderma* teeth appears to simply be another deeper dentine infolding. Furthermore, despite the fact that the venom gland is only present in the lower jaw in *Heloderma* [2], plicidentine and a distinct groove also develop in all of the maxillary teeth, underscoring the disconnection between tooth grooves and venom glands (electronic supplementary material, figure S3c–e). Contrary to the previous study [16] arguing that dentine infoldings in 'varanoid' lizards and snakes differ in the way they develop, we observed no major differences in the development of the plicidentine in *Varanus* and the elapids *H. cyanocinctus* and *O. scutellatus* beyond the degree to which the dentine is folded (electronic supplementary material, text and figure S8).

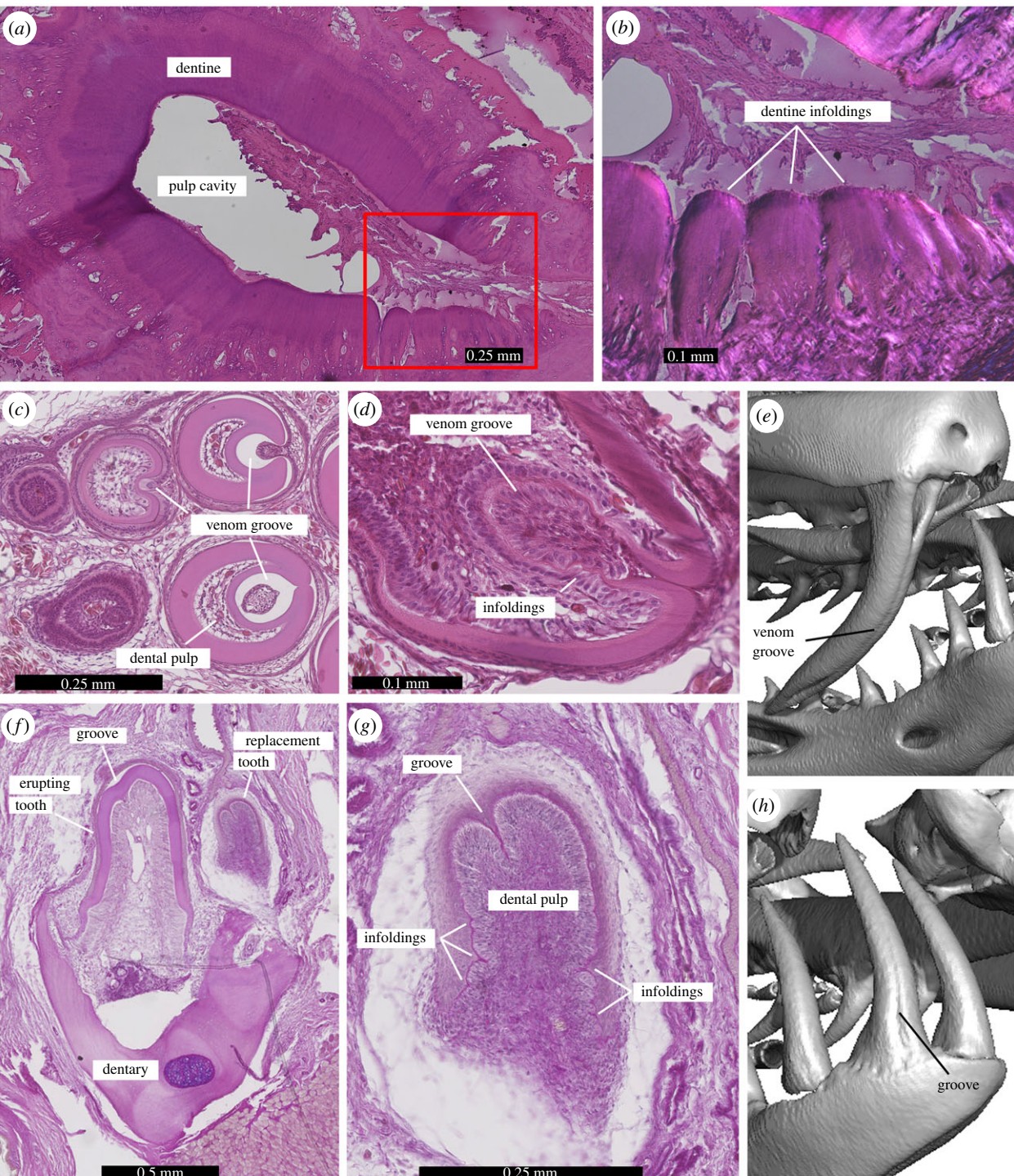

**Figure 2.** Venom grooves have the same developmental origin as plicidentine infoldings. (*a*) Horizontal section (haematoxylin and eosin) through the base of a *Boa constrictor* maxillary tooth showing infolded dentine (plicidentine) at the front (anterior to the right). (*b*) Close-up of the plicidentine infoldings in (*a*) (red box) under cross-polarized light. (*c*) Cross-sections through growth series of developing fangs in *Hydrophis cyanocinctus* (haematoxylin and eosin). Earlier developmental stages to the left and/or top. (*d*) Close-up of an early developing fang of *H. cyanocinctus* showing plicidentine infoldings in close association with the developing venom groove. (*e*) Fully developed venom fang in *Oxyuranus scutellatus*. The venom groove is closed by a suture along the mid-portion of the tooth, but remains open dorsally and ventrally. (*f*) Plicidentine in developing dentary teeth of *H. cyanocinctus* (haematoxylin and eosin). (*g*) Close-up of the replacement tooth shown in (*f*). Note the presence of a large invagination that will result in a shallow groove in the fully erupted tooth. The same invagination is also visible in the more fully developed dentary tooth in (*f*). (*h*) Fully developed dentary tooth of *O. scutellatus* showing the presence of an anterior shallow groove. (Online version in colour.)

We further sought to test whether a proposed biomechanical role of plicidentine [14] could explain its prevalence across Serpentes. According to a previous review of the possible functions of plicidentine [14], broad plicidentine infoldings may increase resistance to compressive forces and bending moments (i.e. strength and bending resistance) of the teeth. However, up until now, these hypotheses had not been rigorously tested.

We found only minimal differences, if any, in the distribution of strains, which contradicts previous suggestions that the folds may increase the bending resistance of the teeth [14] (figure 4; electronic supplementary material, figure S11). We also found no difference in Von Mises stress magnitudes between the models with and without plicidentine (electronic supplementary material, figures S12 and

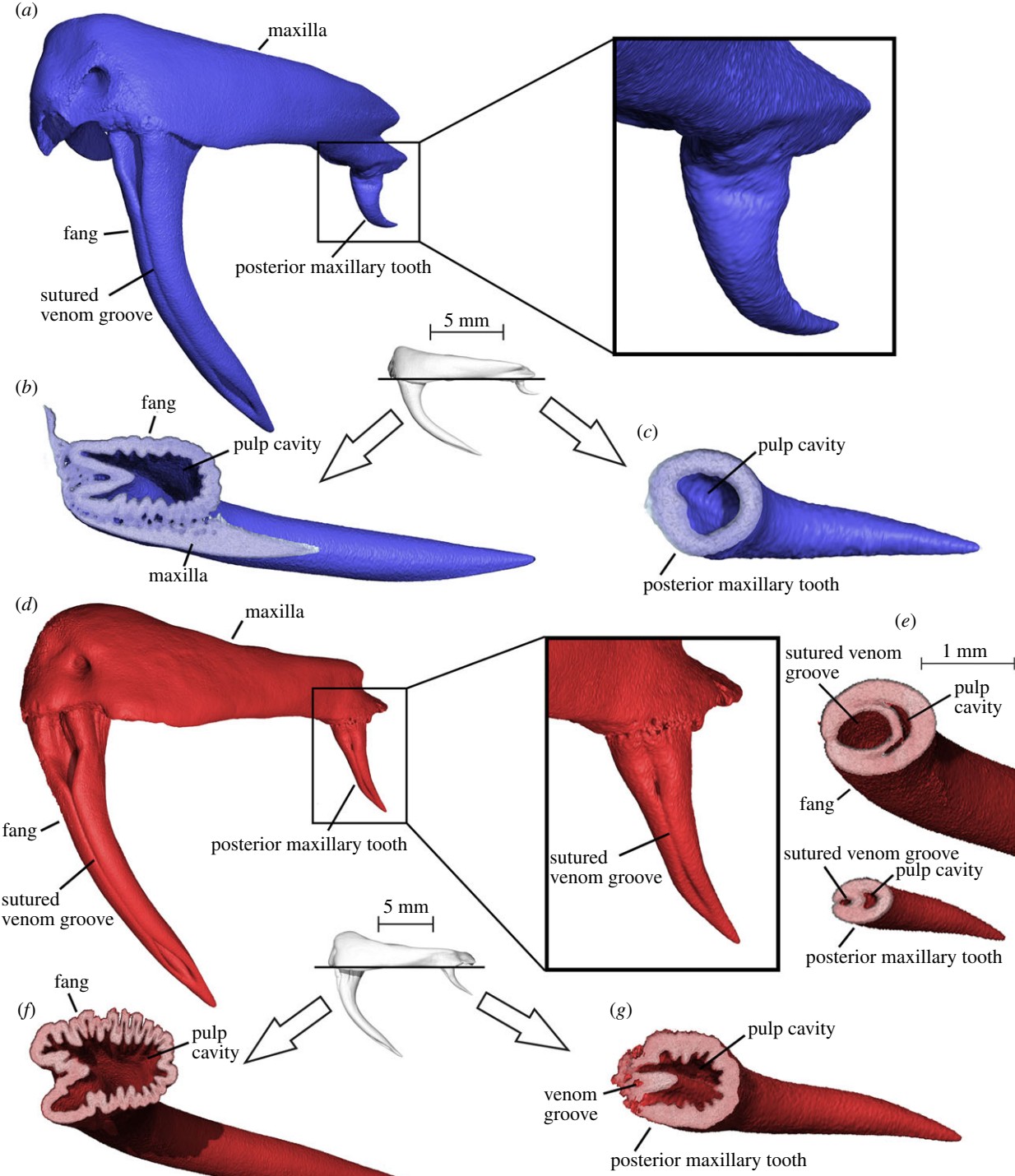

**Figure 3.** Increased expression of plicidentine infolding in a variant individual of *Oxyuranus scutellatus* is associated with the development of an accessory venom groove on the posterior maxillary teeth. (*a*) Anterolateral view of left maxilla of normal specimen of *O. scutellatus*. (*b*,*c*) Horizontal section across the bases of the maxillary teeth (see inset) to expose the plicidentine in the fang (*b*) of the normal specimen of *O. scutellatus* and its absence in the posterior maxillary tooth (*c*). (*d*) Anterolateral view of right maxilla of the variant of *O. scutellatus*. Note how a venom groove has also developed on the posterior maxillary tooth. (*e*) Horizontal sections through the middle of the fang (top) and posterior maxillary tooth (bottom) to show the development of a sutured venom groove (venom duct) in the latter as well. (*f*,*g*) Horizontal section across the bases of the maxillary teeth (see inset) of the variant of *O. scutellatus*. Note how this specimen shows more extensive infolding of the dentine at the base of the fang (folds are more numerous and deeper) (*f*) and that distinct infoldings are associated with the venom groove that developed on the posterior maxillary tooth (*g*). (Online version in colour.)

S13), indicating a similar response to compressive loading (i.e. similar strength).

Similarly, our results found no support for a biomechanical role of the groove during biting either, as we found no major differences in strain or stress regimes between the models with and without a groove (figure 4; electronic supplementary material, figures S11–S13).

## 4. Discussion

The evolution of snake fangs provides an elegant example of how a shared ancestral feature of snake teeth, plicidentine, has been co-opted to repeatedly evolve a new, complex structure able to administer venom. Until this study, it was thought that plicidentine was almost entirely absent in

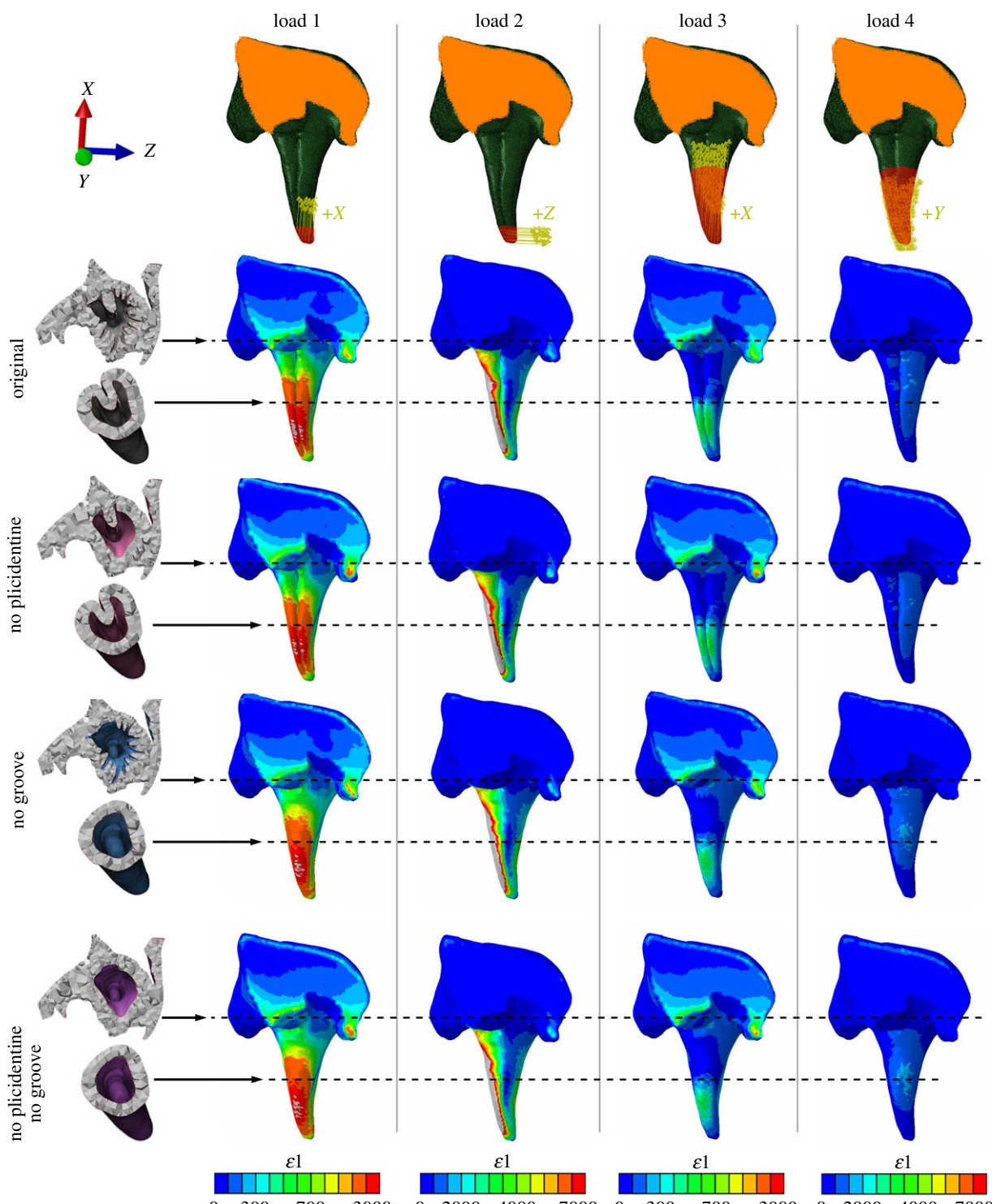

**Figure 4.** FEA results for strain distributions showing maxillae of *F. leucobalia* in anterior view (maxilla cut anteriorly). Comparison is between the original fang model and modified versions which involved removal of the basal plicidentine infoldings (no plicidentine model), the venom groove (no groove model) or both (no plicidentine, no groove model). Boundary conditions for all models are shown at the top (orange area indicates constraint, yellow arrows indicate load direction, red area indicates surface to which the load was applied). Horizontal sections for all models are shown to the left. Colour maps for each loading case are shown at the bottom and represent microstrains ($\mu\varepsilon$) (grey values are beyond scale). Posterior views of the fangs are shown in the electronic supplementary material, figure S11.

modern reptiles, except in some 'varanoid' lizards [12,13,16]. Indeed, most snake teeth show little external evidence of plicidentine. However, high-resolution microCT scanning and histological sections reveal that plicidentine is broadly distributed across the snake phylogeny (figure 5; electronic supplementary material, figures S2 and S3). This feature is ubiquitous and thus ancestrally present in at least alethinophidians, and potentially all snakes (see below).

We provide the first definitive evidence that plicidentine is indeed widespread across snakes (both venomous and non-venomous) and forms the basis for the independent acquisition of venom grooves in colubroid snakes. The repeated origin of very similar venom fangs across snakes is due to the elaboration of a shared fundamental feature of snake dental evolution and development—the ability to produce plicidentine.

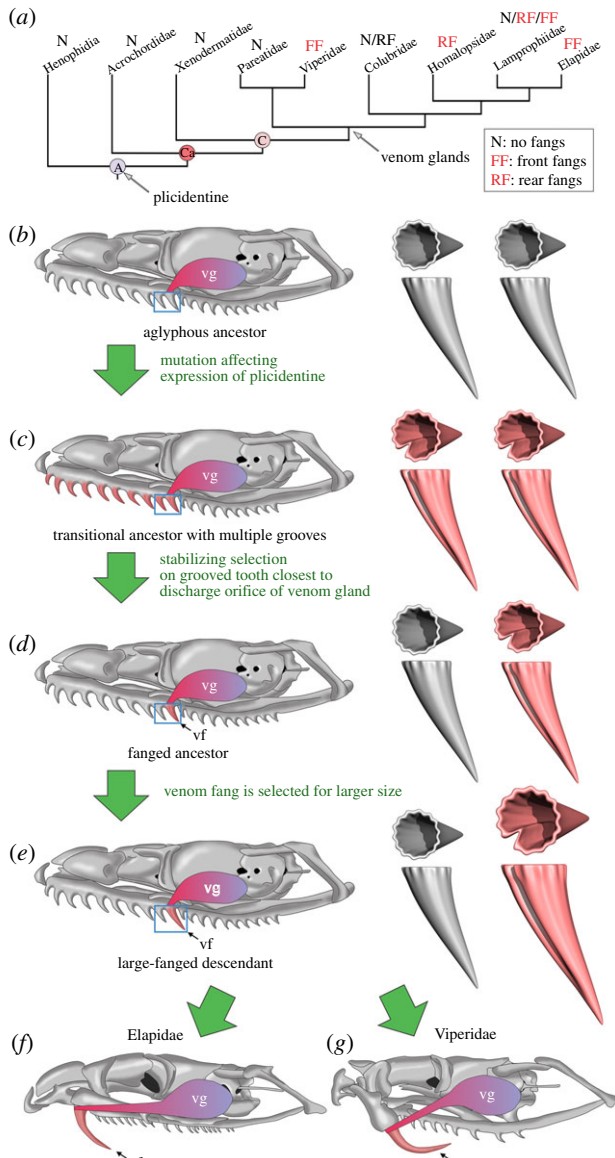

**Figure 5.** The proposed evolutionary hypothesis for the origin of snake venom fangs. (*a*) Distribution of plicidentine, venom glands and types of fangs on a phylogeny of alethinophidian snakes. Phylogenetic relationships and data on the distribution of venom glands and types of fangs are from previous studies [1,3,6,36]. The repeated independent evolution of front and rear venom fangs in snakes can be explained by modifications to tooth development involving the plesiomorphic dentine infoldings (plicidentine) common to all Alethinophidia (*b*–*e*). (*b*) Aglyphous colubroid ancestor possessing both plicidentine (plesiomorphic in Alethinophidia) and a venom gland (plesiomorphic in Colubroidea). (*c*) Appearance of grooves (deeper plicidentine infoldings) in some lineages of colubroids. (*d*) Grooved teeth adjacent to the discharge orifice of the venom gland are subject to stabilizing selection and fixed in several lineages of venomous snakes. (*e*) Venom fangs are further refined (e.g. appearance of tubular fangs) and selected for larger size to increase efficiency in envenomation. (*f,g*) Finally independent antero-posterior shortening of the maxilla and reduction in the number of teeth led to further modifications in some lineages (e.g. Elapidae and Viperidae). Blue box on the skull diagram highlights types of teeth present at each evolutionary stage (tooth type schematics shown on the right). Grooved (and tubular) teeth are highlighted red. Abbreviations: A, Alethinophidia; Ca, Caenophidia; C, Colubroidea; vf, venom fang; vg, venom gland.

The developmental evidence presented here agrees with previous findings [11] that the venom groove originates as an infolding within the epithelial–mesenchyme interface of

the tooth from which dentine and enamel are formed, but provides critical evolutionary context for the origin of this structure. The development of the venom grooves and canals [11] is identical to the development of plicidentine [12]. In both cases, the inner enamel epithelium of a developing tooth folds inwards prior to the formation of mineralized dentine (figure 2*c,d*; electronic supplementary material, figures S4, S5, S7 and S8). In the case of plicidentine, the folds can form all around the tooth base and can be loose undulations and loops, or tight folds with no intervening spaces (figure 2*g*) [12,18]. In the formation of a venom groove or canal (when the groove is fully enclosed), the fold is much larger and more sinuous, forming a loop that occupies the centre of the tooth (figure 2*c,d*) [11]. Dentine deposition only occurs after the epithelial folding is complete in both cases (electronic supplementary material, figure S8). Therefore, venom grooves are simply an elaboration of plicidentine, developing from the same epithelial infolding.

Importantly, *Heloderma* and the taipan variants show that the development of venom grooves can be independent from the presence of a closely associated venom gland or duct. In fact, in some rear-fanged colubroids, the duct of the Duvernoy's gland (the most primitive colubroid venom gland [1,38]) opens directly into the oral cavity rather than to the lumen of the fang sheath and the surface of the fang [1]. Additionally, several colubroids lack grooved teeth despite possessing a Duvernoy's gland [38], which further supports a degree of developmental independence of the two.

The occasional presence of grooves in teeth other than fangs of modern snakes, combined with a close association of venom grooves and plicidentine, not only in snakes but also in both extant (*Heloderma* [2]) and extinct lizards (*Estesia* [39]), indicates that the evolution of a venom groove in squamate reptiles is contingent on the presence of plicidentine and later on a connection to a venom gland.

Because snakes primitively lacked venom delivery systems (i.e. fangs and associated venom glands), which are only found in highly derived snake clades [3], the presence of plicidentine in nearly all snakes, including basal forms such as *Yurlunggur*, raises the question of its original function. We show that, contrary to previous assumptions about the biomechanical significance of plicidentine [14], the infoldings do not have a role in increasing resistance to bending or compression during biting (figure 4; electronic supplementary material, figures S11–S13). These results eliminate all but one of the proposed hypotheses: plicidentine improves the attachment of elongate but shallowly implanted teeth [14]. Basal infoldings would facilitate attachment of the replacement teeth to the dentigerous bone by providing an increased surface area for attachment of the periodontal ligament, which later calcifies in most snakes [37]. We propose that snakes and 'varanoid' lizards share teeth that are, relative to other squamates, relatively tall, slender and with little bony support at their bases; the increased area of attachment provided by plicidentine might be the evolutionary solution for this potential weakness [14].

We also found in our FEA that the further elaboration of a fold into a large groove similarly lacked a stress-release function and did not increase bending resistance of the fang. Therefore, its selection is likely linked to its role in facilitating venom delivery.

Regardless of its original function in snakes, our hypothesis for the origins of the venom groove from a plicidentine infolding

provides a novel and simple explanation for the striking morphological and developmental similarities of snake fangs [4,10] and for the ease with which snake lineages independently evolved venom fangs [1,3,36] (figure 5). The ancestral condition for venomous snakes could have been a random mutation affecting plicidentine expression where several (if not all) maxillary teeth developed a groove (an analogous condition is seen in *Heloderma* and is occasionally observed in extant snakes [34,35]). Later only the groove on the tooth (or teeth [3]), most closely associated with the discharge orifice of the duct connected to the primordial venom gland (Duvernoy's gland [1,38]) was refined due to its adaptive value in facilitating the injection of venom into prey. This tooth must have been located somewhere on the posterior maxillary dental lamina, the development of which is linked to that of the venom gland and its duct [25], but was not necessarily the most posterior tooth in every snake lineage. Therefore, the grooved venom fangs of snakes did not evolve independently and multiple times all across colubroids as completely new structures, but rather resulted from the elaboration and co-option of a previously unrecognized ancestral dental feature common to most snakes.

Data accessibility. High-resolution computed tomography (microCT) scans are freely available from www.morphosource.org (project: Teeth of Serpentes) [40]. FEA models are freely available from figshare.com (doi:10.6084/m9.figshare.14747928) [41].

Authors' contributions. A.P.: conceptualization, data curation, formal analysis, funding acquisition, investigation, methodology, project administration, visualization, writing-original draft, writing-review and editing; A.R.H.L.: formal analysis, funding acquisition, investigation, methodology, visualization, writing-review and editing; O.P.: data curation, formal analysis, investigation, methodology, software, visualization, writing-review and editing; S.G.C.C.: formal analysis, methodology, visualization, writing-review and editing; H.MA.: formal analysis, methodology, visualization, writing-review and editing; M.N.H.: writing-review and editing; A.R.E.: visualization, writing-review and editing; M.W.C.: funding acquisition, writing-review and editing; M.S.Y.L.: funding acquisition, software, supervision, writing-review and editing

All authors gave final approval for publication and agreed to be held accountable for the work performed therein.

Competing interests. Authors declare no competing interests.

Funding. A.R.H.L. was supported by an NSERC Postdoctoral Fellowship. A.P. and M.S.Y.L. were supported by the Australian Research Council (grant no. DP200102328); A.P. was also supported by a Visiting Professor Fellowship from the University of Alberta Faculty of Science. M.W.C. was supported by an NSERC Discovery Grant (#23458) and a University of Alberta Biological Sciences Chair's Research Allowance.

Acknowledgements. We thank A. Oatway for assistance with thin sectioning and staining protocols at the University of Alberta; we thank O. Vernygora and Y. Wong for assistance with microCT scanning and M. Gingras for the use of the microCT scanner at the University of Alberta. We thank Adelaide Microscopy and Microscopy Australia for access to the microCT scanning equipment at the University of Adelaide (Adelaide, Australia), and R. Williams for the assistance provided; K. Batra, A. Labridinis and E. Schneider from Histology Services at the Faculty of Health and Medical Sciences, University of Adelaide; For assistance in museum collections, we wish to thank D. Kizirian, R. Pascocello and M. G. Arnold (AMNH).

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
