## [Peer Review File · Proceedings of the Royal Society B: Biological Sciences]

Review History

RSPB-2021-0453.R0 (Original submission)

Review form: Reviewer 1 (Anthony Herrel)

Recommendation

Major revision is needed (please make suggestions in comments)

Scientific importance: Is the manuscript an original and important contribution to its field?

Good

General interest: Is the paper of sufficient general interest?

Good

Quality of the paper: Is the overall quality of the paper suitable?

Good

Is the length of the paper justified?

Yes

Should the paper be seen by a specialist statistical reviewer?

No

Do you have any concerns about statistical analyses in this paper? If so, please specify them explicitly in your report.

No

It is a condition of publication that authors make their supporting data, code and materials available - either as supplementary material or hosted in an external repository. Please rate, if applicable, the supporting data on the following criteria.

Is it accessible?

Yes

Is it clear?

Yes

Is it adequate?

No

Do you have any ethical concerns with this paper?

No

Comments to the Author

In the manuscript the authors study the evolution of snake fangs using micro CT imaging and biomechanical analysis. They show that many snakes have an infolding of the dentine at the base of the tooth which is a likely precursor for the venom groove. The data suggest that these grooves mechanically reinforce the teeth and consequently might be an evolutionary precursor for the evolution of fangs. These results are novel and of interest and should appeal to a broad audience. I enjoyed reading this paper and think that it would make a great paper for Proceedings B. However I have a few major and some minor comments I would like to see addressed in a revision.

Major: the authors try to spin the paper by arguing that snake fangs did not evolve repeatedly de novo but rather are the result of a co-option of existing features of the teeth. I totally agree with the statement but don't think anyone ever really claimed that these were the result of 'de novo' evolution. This is a largely semantic debate obviously as by definition an organism can only produce traits based on the genetic toolbox available ... as such true 'de novo' traits are by definition nearly inexistent (unless you resurrect the hopeful monster theory). All this just to say that the results are original and interesting and that you do not need to try to oversell the paper.

On line 82 the authors argue that infolding is super important as it increases mechanical resistance. The latter is indeed suggested by the models, but anyone who has been bitten by a few snakes or has seen snakes feed on large prey knows that teeth break all the time, but are quickly replaced. So how important is it really to avoid breakage? I'd like to see this briefly discussed in the paper.

line 104: Fordonia is about the worst possible species to choose as a comparison to Oxyuranus... Fordonia is hyper specialised in eating hard shelled crabs. It bites harder than any snake and than most lizards I measured and I've seen (video) it crush hard shelled crabs. So yes, for Fordonia avoiding the teeth from breaking is likely quite important... why not use another species that is a little less specialised? This would allow you to make the argument about the role of plicidentine in reinforcing snake teeth and the importance thereof. As it stands what I take home is that in an exceptional snake like Fordonia yes plicidentine is important. However, this does not really provide support for the generalized role of plicidentine in reinforcing teeth and thus its importance as an evolutionary precursor of fangs.

line 134: why use different Young's moduli for the two species. This makes no sense ... the 18GPA seems more reasonable as it is derived from an actual biomechanics study.

line 138: why 5N ? just because another author used this value ? It would have been so much more useful to estimate the forces based on the muscle cross sectional areas. The models aren't even comparable as you load the Fordonia model with only 0.19 N. yet, in reality this snake has way stronger muscles and the in vivo loads are much higher. I understand if you want to load a model with an arbitrary force for the sake of comparison but here this is not what is done. The consequence is that 1) models are not comparable and 2) models do not reflect in vivo conditions. So what do the models really tell us ? I'm pretty sure you're on the right track, but this really needs to be improved to either make models comparable or make the models ecologically relevant. In a perfect world I would even like to see a third model for a less specialized snake in addition to the durophagous Fordonia.

So whereas I like the paper and think that the authors are onto something I do think that this needs to be improved. Give the importance of the models I would also like to see at least one figure in the main text.

Review form: Reviewer 2

Recommendation

Major revision is needed (please make suggestions in comments)

Scientific importance: Is the manuscript an original and important contribution to its field?

Excellent

General interest: Is the paper of sufficient general interest?

Excellent

Quality of the paper: Is the overall quality of the paper suitable?

Acceptable

Is the length of the paper justified?

Yes

Should the paper be seen by a specialist statistical reviewer?

No

Do you have any concerns about statistical analyses in this paper? If so, please specify them explicitly in your report.

No

It is a condition of publication that authors make their supporting data, code and materials available - either as supplementary material or hosted in an external repository. Please rate, if applicable, the supporting data on the following criteria.

Is it accessible?

Yes

Is it clear?

Yes

Is it adequate?

Yes

Do you have any ethical concerns with this paper?

No

Comments to the Author

See attached comments. (See Appendix A)

Decision letter (RSPB-2021-0453.R0)

19-Mar-2021

Dear Dr Palci:

I am writing to inform you that your manuscript RSPB-2021-0453 entitled "Plicidentine and the repeated origins of snake venom fangs" has, in its current form, been rejected for publication in Proceedings B.

This action has been taken on the advice of referees, who have recommended that substantial revisions are necessary. With this in mind we would be happy to consider a resubmission, provided the comments of the referees are fully addressed. However please note that this is not a provisional acceptance.

There is some dispute over the value of the FEA in the MS and the fairness in how it was set up; (vs. hypotheses) plus why it didn't really feature in the main MS. Please deal with this carefully if you resubmit.

Importantly, in review it was noted that the CT scan datasets are not openly available-- "However, CT scan datasets are available from the corresponding author upon request and with permission from the SAMA." this statement does not fit our policy. The datasets must be made accessible such as via Morphosource, which can protect museums as well as maintain openness.

To upload a resubmitted manuscript, log into <http://mc.manuscriptcentral.com/prsb> and enter your Author Centre, where you will find your manuscript title listed under "Manuscripts with

Decisions." Under "Actions," click on "Create a Resubmission." Please be sure to indicate in your cover letter that it is a resubmission, and supply the previous reference number.

Sincerely,
Dr John Hutchinson, Editor
mailto: proceedingsb@royalsociety.org

Associate Editor
Board Member: 1
Comments to Author:

Thank for the opportunity to review this study. Herein the authors examine the development, mechanics and evolution of the venom delivery apparatus in snakes. My thoughts mirror those the reviewers exactly: this is a really interesting and understudied topic, and the paper is commended for examining it using a nice array of complimentary methods. The paper also yields interesting new insights and I think it carries broad biological interest. I'm also with the reviewers completely in their assessment of the FEA as it currently stands. This is the major weakness and must be addressed. Reviewer 1 also notes the strange absence of FEA data/figures from the main text: I made the same comment to the Editor when I recommended the paper be sent out for review. If (a revised version of) the FEA is to be presented in a resubmission and form a major component of the story (as it did in this version) then I think the reader needs to see the evidence for the interpretations in the main text.

(In addition - a thank you to the two reviewers for providing detailed reports so quickly in these challenging times. It is very much appreciated).

Reviewer(s)' Comments to Author:

Referee: 1

Comments to the Author(s)

In the manuscript the authors study the evolution of snake fangs using micro CT imaging and biomechanical analysis. They show that many snakes have an infolding of the dentine at the base of the tooth which is a likely precursor for the venom groove. The data suggest that these grooves mechanically reinforce the teeth and consequently might be an evolutionary precursor for the evolution of fangs. These results are novel and of interest and should appeal to a broad audience. I enjoyed reading this paper and think that it would make a great paper for Proceedings B. However I have a few major and some minor comments I would like to see addressed in a revision.

Major: the authors try to spin the paper by arguing that snake fangs did not evolve repeatedly de novo but rather are the result of a co-option of existing features of the teeth. I totally agree with the statement but don't think anyone ever really claimed that these were the result of 'de novo' evolution. This is a largely semantic debate obviously as by definition an organism can only produce traits based on the genetic toolbox available ... as such true 'de novo' traits are by definition nearly inexistent (unless you resurrect the hopeful monster theory). All this just to say that the results are original and interesting and that you do not need to try to oversell the paper.

On line 82 the authors argue that infolding is super important as it increases mechanical resistance. The latter is indeed suggested by the models, but anyone who has been bitten by a few snakes or has seen snakes feed on large prey knows that teeth break all the time, but are quickly replaced. So how important is it really to avoid breakage ? I'd like to see this briefly discussed in the paper.

line 104: Fordonia is about the worst possible species to choose as a comparison to Oxyuranus... Fordonia is hyper specialised in eating hard shelled crabs. It bites harder than any snake and than most lizards I measured and I've seen (video) it crush hard shelled crabs. So yes, for Fordonia avoiding the teeth from breaking is likely quite important... why not use another species that is a

little less specialised? This would allow you to make the argument about the role of plicidentine in reinforcing snake teeth and the importance thereof. As it stands what I take home is that in an exceptional snake like *Fordonia* yes plicidentine is important. However, this does not really provide support for the generalized role of plicidentine in reinforcing teeth and thus its importance as an evolutionary precursor of fangs.

line 134: why use different Young's moduli for the two species. This makes no sense ... the 18GPA seems more reasonable as it is derived from an actual biomechanics study.

line 138: why 5N ? just because another author used this value ? It would have been so much more useful to estimate the forces based on the muscle cross sectional areas. The models aren't even comparable as you load the *Fordonia* model with only 0.19 N. yet, in reality this snake has way stronger muscles and the in vivo loads are much higher. I understand if you want to load a model with an arbitrary force for the sake of comparison but here this is not what is done. The consequence is that 1) models are not comparable and 2) models do not reflect in vivo conditions. So what do the models really tell us ? I'm pretty sure you're on the right track, but this really needs to be improved to either make models comparable or make the models ecologically relevant. In a perfect world I would even like to see a third model for a less specialized snake in addition to the durophagous *Fordonia*.

So whereas I like the paper and think that the authors are onto something I do think that this needs to be improved. Give the importance of the models I would also like to see at least one figure in the main text.

Referee: 2

Comments to the Author(s)
See attached comments

Author's Response to Decision Letter for (RSPB-2021-0453.R0)

See Appendix B.

RSPB-2021-1391.R0

Review form: Reviewer 1

Recommendation

Accept with minor revision (please list in comments)

Scientific importance: Is the manuscript an original and important contribution to its field?

Good

General interest: Is the paper of sufficient general interest?

Good

Quality of the paper: Is the overall quality of the paper suitable?

Good

Is the length of the paper justified?

Yes

Should the paper be seen by a specialist statistical reviewer?

No

Do you have any concerns about statistical analyses in this paper? If so, please specify them explicitly in your report.

No

It is a condition of publication that authors make their supporting data, code and materials available - either as supplementary material or hosted in an external repository. Please rate, if applicable, the supporting data on the following criteria.

Is it accessible?

Yes

Is it clear?

Yes

Is it adequate?

Yes

Do you have any ethical concerns with this paper?

No

Comments to the Author

I have now read the revised version of the manuscript entitled 'Plicidentine and the repeated origins of snake venom fangs' by Palci and co-authors. I found the manuscript much improved relative to the initial version and have only some minor comments remaining.

My only real comment is that we have o idea of the size or age of the specimens used. The authors regularly refer to tooth development when they refer to specimens examined but do they refer to the initial development of the tooth in the embryo or the development of the tooth after replacement in an adult. I presume the latter as having access to embryos is difficult, but this was not entirely clear. If all specimens examined were adults then a simple statement in the methods would suffice.

line 208: to the base of the teeth ?

line 295: variants ?

Review form: Reviewer 2

Recommendation

Accept with minor revision (please list in comments)

Scientific importance: Is the manuscript an original and important contribution to its field?

Excellent

General interest: Is the paper of sufficient general interest?

Excellent

Quality of the paper: Is the overall quality of the paper suitable?

Good

Is the length of the paper justified?

Yes

Should the paper be seen by a specialist statistical reviewer?

No

Do you have any concerns about statistical analyses in this paper? If so, please specify them explicitly in your report.

No

It is a condition of publication that authors make their supporting data, code and materials available - either as supplementary material or hosted in an external repository. Please rate, if applicable, the supporting data on the following criteria.

Is it accessible?

Yes

Is it clear?

N/A

Is it adequate?

Yes

Do you have any ethical concerns with this paper?

No

Comments to the Author

Please see attached file. (See Appendix C)

Decision letter (RSPB-2021-1391.R0)

08-Jul-2021

Dear Dr Palci

I am pleased to inform you that your manuscript RSPB-2021-1391 entitled "Plicidentine and the repeated origins of snake venom fangs" has been accepted for publication in Proceedings B. Congratulations!!

The referee(s) have recommended publication, but also suggest some minor revisions to your manuscript. Therefore, I invite you to respond to the referee(s)' comments and revise your manuscript. Because the schedule for publication is very tight, it is a condition of publication that you submit the revised version of your manuscript within 7 days. If you do not think you will be able to meet this date please let us know.

To revise your manuscript, log into <https://mc.manuscriptcentral.com/prsb> and enter your Author Centre, where you will find your manuscript title listed under "Manuscripts with Decisions." Under "Actions," click on "Create a Revision." Your manuscript number has been appended to denote a revision. You will be unable to make your revisions on the originally

submitted version of the manuscript. Instead, revise your manuscript and upload a new version through your Author Centre.

Sincerely,

Dr John Hutchinson, Editor

Associate Editor

Board Member

Comments to Author:

Thank you for implementing the suggestions from the first round of peer review. The FE has definitely improved and the results are now communicated appropriately in the main text with the additional figure. While the result has changed, I don't see this as a negative - the findings remain novel and interesting. I really like Figure 5. The reviewers have some additional minor issues, which I again I agree with (particularly those of reviewer 2 regarding unnecessarily overstating conclusions, and being too absolute with some of the wording).

Reviewer(s)' Comments to Author:

Referee: 1

Comments to the Author(s).

I have now read the revised version of the manuscript entitled 'Plicidentine and the repeated origins of snake venom fangs' by Palci and co-authors. I found the manuscript much improved relative to the initial version and have only some minor comments remaining.

My only real comment is that we have o idea of the size or age of the specimens used. The authors regularly refer to tooth development when they refer to specimens examined but do they refer to the initial development of the tooth in the embryo or the development of the tooth after replacement in an adult. I presume the latter as having access to embryos is difficult, but this was not entirely clear. If all specimens examined were adults then a simple statement in the methods would suffice.

line 208: to the base of the teeth ?

line 295: variants ?

Referee: 2

Comments to the Author(s).

Please see attached file

Author's Response to Decision Letter for (RSPB-2021-1391.R0)

See Appendix D.

Decision letter (RSPB-2021-1391.R1)

19-Jul-2021

Dear Dr Palci

I am pleased to inform you that your manuscript entitled "Plicidentine and the repeated origins of snake venom fangs" has been accepted for publication in Proceedings B.

Data Accessibility section

Open Access

Paper charges

Sincerely,

Proceedings B

Appendix A

This is a review for the manuscript “Plicidentine and the repeated origins of snake venom fangs”. The paper sets out to look for plicidentine in a diversity of reptiles, focusing on snakes, and to determine if this dentine in-folding is correlated to venom grooved. There are three main approaches that the paper takes: μ CT scanning, histology, and FEA to test a biomechanical hypothesis regarding the function of plicidentine. The paper nicely demonstrates that a number of reptiles, including all but one species of snake in the study, do have plicidentine at the base of their teeth, and that it likely serves as the starting point for the venom groove/tube in venomous snakes. Based on their observations that grooves develop in teeth that are not associated with Duvernoy’s gland or venom glands, paired with the FEA output, the authors propose a model for venom fang evolution.

I really enjoyed this paper and think that there’s a great deal of fantastic information here. The only portion of the paper that concerns me is the FEA analysis. The comparison between the fang with and without the plicidentine ridges is a great idea, but the method used to smooth those ridges is changing both the volume of the fang and the surface area. From my understanding of FEA comparisons (based on the Dumont et al 2009 – Requirements for comparing performance of finite element models of biological structures), if the volume/surface area is different between the two models then the applied force needs to be scaled to allow for a direct comparison. I worry that the differences being observed may be an artifact of a difference in load. I have similar concerns regarding the groove/no groove models. Finally, I think that there’s a lost opportunity for an informative comparison here. They authors interpret the lack of difference in Von Mises stress magnitude/concentration in the groove/no groove model and the presence of this difference in the plicidentine/no plicidentine model as evidence supporting their proposed model for venom fang evolution. The problem is the loading regimes in these two fangs are very different. Similarly, there are two different sets of material properties assigned to the models, which will potentially affect how the teeth respond to loads, making it more difficult to compare the models. I think, assuming the force being applied is equivalent and appropriate, that this argument would be much better made if the grooved fang were also modeled with and without plicidentine. It seems like it would more immediately test the hypothesis that the plicidentine is adding a structural support and that the venom groove is functionally separate.

While I think the μ CT and histology data on their own still make this a really strong manuscript, I’d love to see the functional aspect more thoroughly fleshed out.

Specific comments follow:

Line 33 (and elsewhere): Be more specific about what aspect of biomechanical performance you’re talking about

Methods – In general a table outlining the species included in the study, which aspect of the study they were included in (in S Table 2), and (especially) what type of tooth/fang they have

would be really informative and/or a phylogenetic tree along the lines of the tree in fig S4 to show how they relate to each other.

Line 89: Was the right side the only side looked at for all specimens in the study?

FEA – most of my concerns are outlined above

Lines 128-131: with such different numbers of elements, was anything done to make sure the results are showing comparable results?

Lines 136-157: It would be much more informative if the figures actually showing the loads were references here – Figs S13, S15, and S16.

Lines 165-166: Why not include histology from non-elapid venomous snakes as well? Given how closely related *Hydrophis* and *Oxyuranus* are compared to the other species included in the study it seems important to verify that the patterns you're seeing are wide-spread phylogenetically

Line 191: from my notes: COOL!

Line 193: S Table 1 seems like more of a methods table

Lines 219-222: While it's in the supplemental text, I feel like a brief discussion of how the teeth form would go a long way to putting this in context and lending it more weight.

Lines 223-227: from my notes: COOL!

Lines 227-231: This is such an interesting finding, and I think really needs more emphasis. This is the line of evidence that I found the most compelling in thinking about the proposed evolution of the venom groove/tube.

Lines 241- 247: from my notes: Do you see this in the Heloderma/ {arrow down to next paragraph} There we go!

Lines 268 – 270: the load simulating the fully embedded tooth seems like the more structurally/functionally relevant piece of info from the perspective of the FEA analysis.

Line 271: refers to fig S12b, but that doesn't seem applicable – may be a typo?

Line 272: “middle of the crow” I don't know why but this description threw me off. It might be more intuitive to refer to that location as mid-fang length?

Line 282 – 285: Looking at the figures, it does look like there is a noticeable difference in the Von Mises stress between panels h and i. Granted it's not as large as the change seen in the other fang, but the overall magnitude and range are much smaller as well. It seem notable that this load and the load that seems to show the difference in the other fang are both putting a similar bending load on the fangs.

Figures:

Fig 1: the red/blue doesn't add to the figure, and actually makes it harder to see in BW. That said, the white used in panels a and b really highlights the teeth. I would suggest using that to instead show-off the infolding in the fangs

Fig 3: again, the colors don't help and make it harder to see in BW. Overall the figure is a bit busy, and could probably be cleaned up by not having the duplicate maxillae. Instead a larger version of the light maxilla with the section shown should work. It would also be helpful to see where the additional sections in e were taken

Fig 4: This is a very pretty figure. Would it add anything to add another level below e showing the independent evolution of the 4 types of venomous fang?

Supplemental Material

S Text: is this referred to in the main text at all?

S Fig 4: I really like the idea behind this figure and would almost like to see it in the main text. I like that the rear fangs and front fangs are separated out, though it might be more informative to label grooved fangs, and the different types of front fangs. The sections through the teeth are a little too small to see clearly, and it might be better to do without the skulls? Finally in the phylogeny theres a typo: "*Vipera Berus*"

S. Fig 5: Again a really informative figure, but might be more informative with cartoons like in S Fig 4. Also why are there sections through different levels of the *Bitis gabonica* fang?

S Fig 6: It might help comparison between the μ CT reconstruction and the histology if the μ CT portion was zoomed-in on the base of the fang.

S Fig 8: from my notes: NICE! Also you should probably define pl and de in the legend.

S Fig 9: from my notes: NICE

Appendix B

Please find our replies to the editors and referees' comments highlighted in blue font below.

Editor

In review it was noted that the CT scan datasets are not openly available-- "However, CT scan datasets are available from the corresponding author upon request and with permission from the SAMA." this statement does not fit our policy. The datasets must be made accessible such as via Morphosource, which can protect museums as well as maintain openness.

We have now uploaded all the CT scans mentioned in our manuscript to Morphosource, as required by the journal policy. The scans have been uploaded under the project title: Teeth of Serpentes (currently set to private, it will be made public pending acceptance of the manuscript; temporary private link for review purposes:

<https://datadryad.org/stash/share/4q3qVs6vTH4PXQw1LCtzI-xVwN2siMof2UTbP97fOYI>).

We have also uploaded all of our FEA models to Figshare:

<https://figshare.com/account/home#/projects/114543> (Figshare project link will be made public pending acceptance of the manuscript; temporary link for review purposes:

<https://figshare.com/s/01cd8e4519f81e26ae51>). This information has been added under Data accessibility at the end of our manuscript.

Associate Editor

Board Member: 1

Comments to Author:

Thank for the opportunity to review this study. Herein the authors examine the development, mechanics and evolution of the venom delivery apparatus in snakes. My thoughts mirror those the reviewers exactly: this is a really interesting and understudied topic, and the paper is commended for examining it using a nice array of complimentary methods. The paper also yields interesting new insights and I think it carries broad biological interest. I'm also with the reviewers completely in their assessment of the FEA as it currently stands. This is the major weakness and must be addressed. Reviewer 1 also notes the strange absence of FEA data/figures from the main text: I made the same comment to the Editor when I recommended the paper be sent out for review. If (a revised version of) the FEA is to be presented in a resubmission and form a major component of the story (as it did in this version) then I think the reader needs to see the evidence for the interpretations in the main text.

We have updated all FEMs and have now added an FEA figure to the main text in addition to our figures in the supplementary information.

Reviewer(s)' Comments to Author:

Referee: 1

Comments to the Author(s):

Major: the authors try to spin the paper by arguing that snake fangs did not evolve repeatedly de novo but rather are the result of a co-option of existing features of the teeth. I totally agree

with the statement but don't think anyone ever really claimed that these were the result of 'de novo' evolution. This is a largely semantic debate obviously as by definition an organism can only produce traits based on the genetic toolbox available ... as such true 'de novo' traits are by definition nearly inexistent (unless you resurrect the hopeful monster theory). All this just to say that the results are original and interesting and that you do not need to try to oversell the paper.

We agree that the term “de novo” may be misleading, we have now removed it from our manuscript.

On line 82 the authors argue that infolding is super important as it increases mechanical resistance. The latter is indeed suggested by the models, but anyone who has been bitten by a few snakes or has seen snakes feed on large prey knows that teeth break all the time, but are quickly replaced. So how important is it really to avoid breakage ? I'd like to see this briefly discussed in the paper.

Following the reviewers' suggestions, we ran a new set of models using the *Fordonia* fang coupled with better controlled forces and loading cases. Forces are now scaled by surface area for each model when comparing stresses and by volume when comparing strains, as per the recommendations in Dumont et al. (2009) (now cited as ref. 29). Our new results no longer support the functional role of the plicidentine during biting. The small differences between models cannot justify plicidentine's functional significance in increasing resistance to compression or bending (contra ref. 14). This negative result is an important discovery and its implications are now discussed on page 13 lines 307-322.

line 104: *Fordonia* is about the worst possible species to choose as a comparison to *Oxyuranus*... *Fordonia* is hyper specialised in eating hard shelled crabs. It bites harder than any snake and than most lizards I measured and I've seen (video) it crush hard shelled crabs. So yes, for *Fordonia* avoiding the teeth from breaking is likely quite important... why not use another species that is a little less specialised? This would allow you to make the argument about the role of plicidentine in reinforcing snake teeth and the importance thereof. As it stands what I take home is that in an exceptional snake like *Fordonia* yes plicidentine is important. However, this does not really provide support for the generalized role of plicidentine in reinforcing teeth and thus its importance as an evolutionary precursor of fangs.

The comparison was not between *Oxyuranus* and *Fordonia*, but between teeth of the same snake species with or without plicidentine or with and without venom groove. We understand that the choice of two species might confuse readers. To make things less confusing and more consistent, we now have run all FEAs on models derived from the same fang (same snake, *Fordonia*) after removing the plicidentine, the groove, or both. Contrary to what the reviewer suggested, *Fordonia* is arguably the best snake to test for a possible biomechanical role of plicidentine during biting. Our purpose was to first establish if there was any biomechanical significance to the plicidentine (thus we looked in a snake where that purpose would be most likely), before we addressed whether that function was ubiquitous amongst serpents more generally. Because such role cannot be observed in a durophagous snake, then it will be even

more unlikely to be observed in any other snake. Furthermore, other rear-fanged snakes have smaller-scale dentine infoldings compared to *Fordonia*, which are very hard to model in a 3D mesh meant for FEA without introducing modelling errors. We have now included a paragraph at the start of our section titled “Finite element analysis (FEA)” (in Materials and methods) where we discuss these points (page 5, lines 94-102).

line 134: why use different Young's moduli for the two species. This makes no sense ... the 18GPa seems more reasonable as it is derived from an actual biomechanics study.

As per the comment above, the comparison was within and not between species. For simplification, we have now updated our material properties following the empirical measurements from Van Vuuren et al (2016) and Broeckhoven and Du Plessis (2017). We adopted a Young modulus of 20 GPa, which is the average of the range of values from Van Vuuren et al. (2016); the same value was also used in Broeckhoven and Du Plessis (2017).

line 138: why 5N ? just because another author used this value ? It would have been so much more useful to estimate the forces based on the muscle cross sectional areas. The models aren't even comparable as you load the *Fordonia* model with only 0.19 N. yet, in reality this snake has way stronger muscles and the in vivo loads are much higher. I understand if you want to load a model with an arbitrary force for the sake of comparison but here this is not what is done. The consequence is that 1) models are not comparable and 2) models do not reflect in vivo conditions. So what do the models really tell us ? I'm pretty sure you're on the right track, but this really needs to be improved to either make models comparable or make the models ecologically relevant. In a perfect world I would even like to see a third model for a less specialized snake in addition to the durophagous *Fordonia*.

We apologise for the confusion created by assessing the role of the groove and plicidentine in 2 different species. Our initial comparisons were not between species, rather within species. We do, however, agree with the reviewer that although the volume and surface differences between the model variants are minimal, we should have scaled the force for each model. As explained above, we ran a new set of analyses on four 3D models modified from the same snake fang (original, no groove, no plicidentine, no groove and no plicidentine) where forces are now scaled by model surface when comparing stresses and by model volume when comparing strains, as per the recommendations in Dumont et al. (2009) (now cited as ref. 29).

We also agree that a value of 5N taken from the literature may not reflect the *in vivo* loading conditions. Ideally, we would have liked to have muscle force data to load our FEMs in a more physiological manner. However, such data are not available, and we do not have access to appropriate specimens to experimentally derive these data ourselves. Unfortunately, bite force measurements from snakes are very scant in the literature, and we could only find some data in the work of Penning (2017) (now ref. 28), who studied the colubrid snake *Lampropeltis*. Thus, we estimated the maximum bite force of *Fordonia* following the work of Penning (2017), according to whom head height is the best proxy to infer maximum bite force in a snake. We used the equation from his regression of head height vs maximum bite force, $\log(F) = 1.12 \cdot \log(HH) + 0.47$, to obtain an estimated max bite force value of 5.3 N in our specimen of *Fordonia*. We then scaled this force value to the surface area (for stress

calculations) and to the volume (for strain calculations) as per the recommendations in Dumont et al. (2009). Regarding the comment about testing an additional rear-fanged colubrid, please see our reply above explaining the reasons behind our choice of *Fordonia* (and page 5, lines 94-102 in the main manuscript).

So whereas I like the paper and think that the authors are onto something I do think that this needs to be improved. Give the importance of the models I would also like to see at least one figure in the main text.

We have added the FEA results and one summary figure to the main text (now figure 4).

Referee: 2

Comments to the Author(s):

I really enjoyed this paper and think that there's a great deal of fantastic information here. The only portion of the paper that concerns me is the FEA analysis. The comparison between the fang with and without the plicidentine ridges is a great idea, but the method used to smooth those ridges is changing both the volume of the fang and the surface area. From my understanding of FEA comparisons (based on the Dumont et al 2009 – Requirements for comparing performance of finite element models of biological structures), if the volume/surface area is different between the two models then the applied force needs to be scaled to allow for a direct comparison. I worry that the differences being observed may be an artifact of a difference in load. I have similar concerns regarding the groove/no groove models.

We completely agree with the reviewer. This was an oversight we have now addressed. We now scaled the force based on the surface of the models for stress assessment and based on their volume for strain assessment, as per recommendations in Dumont et al. (2009).

Finally, I think that there's a lost opportunity for an informative comparison here. They authors interpret the lack of difference in Von Mises stress magnitude/concentration in the groove/no groove model and the presence of this difference in the plicidentine/no plicidentine model as evidence supporting their proposed model for venom fang evolution. The problem is the loading regimes in these two fangs are very different. Similarly, there are two different sets of material properties assigned to the models, which will potentially affect how the teeth respond to loads, making it more difficult to compare the models. I think, assuming the force being applied is equivalent and appropriate, that this argument would be much better made if the grooved fang were also modeled with and without plicidentine. It seems like it would more immediately test the hypothesis that the plicidentine is adding a structural support and that the venom groove is functionally separate.

We agree with the reviewer. We have now rerun all FEA on the *Fordonia* fang and on model variants without plicidentine, without groove, and without both plicidentine and groove. We have used a more accurate force estimate than the one adopted in our previous submission (see similar reply to Referee 1 above), and we have scaled the force to the surface (for stress

calculations) and volume (for strain calculations) as per Dumont et al. (2009). We have also kept material properties assignment consistent amongst all models.

While I think the μ CT and histology data on their own still make this a really strong manuscript, I'd love to see the functional aspect more thoroughly fleshed out.

Specific comments follow:

Line 33 (and elsewhere): Be more specific about what aspect of biomechanical performance you're talking about

Noted, we are now specifying that we are referring to biting performance, and in particular to resistance to compression (stresses) and bending (strains).

Methods – In general a table outlining the species included in the study, which aspect of the study they were included in (in S Table 2), and (especially) what type of tooth/fang they have would be really informative and/or a phylogenetic tree along the lines of the tree in fig S4 to show how they relate to each other.

All species (and specimens) used in this study are listed in Tables S1 and S2. We have added some systematic information (family or higher taxon) to make it clear how these species relate to each other. The same families and clades are shown in our figure 5 and figure S2. Presence and position of the fangs for members of these families is indicated both in figure 5 and figure S2.

Line 89: Was the right side the only side looked at for all specimens in the study?

No, only when destructive sampling was necessary we looked only at one side (e.g. *Boa constrictor* histology sections).

FEA – most of my concerns are outlined above

We believe those have now been addressed.

Lines 128-131: with such different numbers of elements, was anything done to make sure the results are showing comparable results?

We did not compare our results between *Oxyuranus* and *Fordonia*, but only within the same snake. In any case, now we only present results for *Fordonia*, following the reviewer's suggestion to have all tests (original, groove absent, plicidentine absent, both groove and plicidentine absent) done on the same fang.

Lines 136-157: It would be much more informative if the figures actually showing the loads were references here – Figs S13, S15, and S16.

Here we have added a reference to our new supplementary figure S1, which shows the loading cases. Images of the loadings for all four cases are now shown also in our figure 4, as well as in our supplementary figures S11-13.

Lines 165-166: Why not include histology from non-elapid venomous snakes as well? Given how closely related *Hydrophis* and *Oxyuranus* are compared to the other species included in the study it seems important to verify that the patterns you're seeing are wide-spread phylogenetically

The plicidentine infoldings can already be observed in our microCT scans of several non-elapid snakes across a very broad phylogenetic sample (supplementary figures S2, S3). Furthermore, histology sections require destructive sampling, so we only did that for a limited number of specimens that were easily available to our lab.

Line 193: S Table 1 seems like more of a methods table

Agreed, we removed the reference to Table S1 from this line.

Lines 219-222: While it's in the supplemental text, I feel like a brief discussion of how the teeth form would go a long way to putting this in context and lending it more weight.

We have transferred here the section describing tooth development from the supplementary text.

Lines 227-231: "In fact, grooved teeth can develop anywhere in the maxilla (see below), and to a lesser degree even in the lower jaw, despite the lack of a connection to a venom gland. The dentary teeth of the examined elapids have prominent infoldings in early stages of tooth development (figure 2f,g), which are associated with a thin groove, or "furrow" [33], situated on the anterior surface of the crowns in mature teeth"

This is such an interesting finding, and I think really needs more emphasis. This is the line of evidence that I found the most compelling in thinking about the proposed evolution of the venom groove/tube.

We elaborate on this finding and its implications in the Discussion (page 14, lines 295-306 and further on page 15 lines 327-329). We have now also added a line to our abstract that reads as follows: "Derivation of the venom groove from a large plicidentine fold that develops early in tooth ontogeny reveals how snake venom fangs could originate repeatedly through the co-option of a pre-existing dental feature even without close association to a venom duct."

Lines 268 – 270: the load simulating the fully embedded tooth seems like the more structurally/functionally relevant piece of info from the perspective of the FEA analysis.

No longer relevant. After rerunning our analyses the performance of the fully embedded tooth is no longer significantly different from those under other loading scenarios. This part has been removed.

Line 271: refers to fig S12b, but that doesn't seem applicable – may be a typo?

No longer relevant, this part has been removed.

Line 272: “middle of the crow” I don’t know why but this description threw me off. It might be more intuitive to refer to that location as mid-fang length?

No longer relevant, this part has been removed.

Line 282 – 285: Looking at the figures, it does look like there is a noticeable difference in the Von Mises stress between panels h and i. Granted it’s not as large as the change seen in the other fang, but the overall magnitude and range are much smaller as well. It seems notable that this load and the load that seems to show the difference in the other fang are both putting a similar bending load on the fangs.

No longer relevant, this figure has been removed and replaced with the results of the new FEA.

Figures:

Fig 1: the red/blue doesn’t add to the figure, and actually makes it harder to see in BW. That said, the white used in panels a and b really highlights the teeth. I would suggest using that to instead show-off the infolding in the fangs

We coloured the fangs red to highlight the fact that they are venom fangs, unlike the teeth shown in (a) and (b), and also to show where they occur in the mouth (skulls). In order to improve the readability of the image in black and white we have decreased the intensity of the red and made it lighter.

Fig 3: again, the colors don’t help and make it harder to see in BW. Overall the figure is a bit busy, and could probably be cleaned up by not having the duplicate maxillae. Instead a larger version of the light maxilla with the section shown should work. It would also be helpful to see where the additional sections in e were taken.

We have adjusted the tone of the colours so that they look a bit lighter in black and white. We don’t think that removing the “duplicate maxilla” would help with the readability of the figure. Also, the “duplicate maxilla” is not a duplicate, but in fact a rendering of the maxilla of each specimen and provides a scale for each. We believe that adding additional lines, arrows, or insets to show where the sections in (e) come from would only make the figure more cluttered. Therefore, we have specified in our figure caption that those are sections through the middle of the fang and posterior maxillary tooth.

Fig 4: This is a very pretty figure. Would it add anything to add another level below e showing the independent evolution of the 4 types of venomous fang?

We have added the skulls of two front-fanged snakes to complete this figure.

Supplemental Material

S Text: is this referred to in the main text at all?

Yes, on what was line 260 in the original submission (now line 250).

S Fig 4: I really like the idea behind this figure and would almost like to see it in the main text. I like that the rear fangs and front fangs are separated out, though it might be more informative to label grooved fangs, and the different types of front fangs. The sections through the teeth are a little too small to see clearly, and it might be better to do without the skulls? Finally in the phylogeny there's a typo: "*Vipera Berus*"

We corrected the typo for *Vipera berus*. We believe that labelling the different types of fangs beyond rear fangs and front fangs would add too much information to this figure, which is meant to provide a quick glimpse at how fangs are distributed across the snake phylogeny. Also, there are no terms to distinguish the front fangs of viperids from those of atractaspidines and elapids, and grooved fangs is synonymous with our use of the term rear fangs (all rear fangs are grooved fangs). We believe that the skulls provide the extra details about type and position of the fangs, so we would like to leave them in. While we would like to include this figure in the main text we also have to comply with space limitations for publication, and by keeping this figure in our supplementary readers can zoom in closely on the various tooth cross sections, thus avoiding the need of making those images larger and removing the skulls.

S. Fig 5: Again a really informative figure, but might be more informative with cartoons like in S Fig 4. Also why are there sections through different levels of the *Bitis gabonica* fang?

This figure was included to present the raw data images (Fig. S4 [now Fig. S2] only show a summary based on interpretative drawings). There are sections through multiple levels of *B. gabonica* to show how the cross section of the fang varies along its length.

S Fig 6: It might help comparison between the μ CT reconstruction and the histology if the μ CT portion was zoomed-in on the base of the fang.

We have added a close-up on the microCT image of the base of the fang.

S Fig 8: from my notes: NICE! Also you should probably define pl and de in the legend.

We have added definitions for the two abbreviations.

Appendix C

I was pleased to re-review the manuscript, “Plicidentine and the repeated origins of snake venom fangs,” and am impressed at the work that has been done to address the major issues raised during the first round of review. I appreciate that the change in the FEA and the resultant change in the output has led to a shift in the discussion, but am worried about how the authors introduce their alternative hypothesis regarding tooth attachment. This seems like a likely scenario, but not necessarily, as stated in the text, the only remaining possibility. This, coupled with some of the wording regarding the hypothesized ancestral condition does still feel like over-selling. That said, it’s a great study and I’m excited to see it published.

Additional Comments:

Abstract:

Lines 28 & 33: “advanced” is a pretty loaded term, and doesn’t really get at the point. Would it be better to say ‘venomous’?

Materials & methods

Line 82: It appears that the authors have replaces “ μ ” throughout the rest of the manuscript with “micro”

FEA section: Much improved overall! The only suggestion I have is that it might be informative here to provide information regarding the scan resolution, even if it can also be found in the supplemental info

Line 126: “anteriorly on the maxilla” makes it sound like only the anterior-most portion of the maxilla is constrained. In looking at the supplemental figure it appears that all of the maxilla immediately underlying the fang is constrained instead. Is there a better way to describe this?

Lines 136-138: This bit may need a bit more explanation. What I’m reading indicates that the Von Mises stresses were only used to compare resistance to compression in grooved vs non-grooved fangs, and that the principal strain data were only used to compare bending in fangs with and without plicidentine?

Discussion

Lines 313-314: “These results leave only one possibility” is a very strong statement, doesn’t account for other possible alternative hypotheses, and implies a conclusion with no data to support it. “These results eliminate all but one of the proposed hypotheses” gets the same information across and doesn’t overstate the findings of the current study.

Supplemental data

S Fig 1: lining up the loads with the different fang treatments is a bit misleading, is there a way to break these apart? Draw a line between the two? Put the loads along the bottom of the fig?

Appendix D

Associate Editor

Board Member

Comments to Author:

Thank you for implementing the suggestions from the first round of peer review. The FE has definitely improved and the results are now communicated appropriately in the main text with the additional figure. While the result has changed, I don't see this as a negative - the findings remain novel and interesting. I really like Figure 5. The reviewers have some additional minor issues, which I again I agree with (particularly those of reviewer 2 regarding unnecessarily overstating conclusions, and being too absolute with some of the wording).

We agree with the suggestions from the Associate Editor and the two reviewers, and have modified our manuscript accordingly (see details below, our replies are highlighted in blue font).

Reviewer(s)' Comments to Author:

Referee: 1

Comments to the Author(s).

I have now read the revised version of the manuscript entitled 'Plicidentine and the repeated origins of snake venom fangs' by Palci and co-authors. I found the manuscript much improved relative to the initial version and have only some minor comments remaining.

My only real comment is that we have no idea of the size or age of the specimens used. The authors regularly refer to tooth development when they refer to specimens examined but do they refer to the initial development of the tooth in the embryo or the development of the tooth after replacement in an adult. I presume the latter as having access to embryos is difficult, but this was not entirely clear. If all specimens examined were adults then a simple statement in the methods would suffice.

We have now added on line 84 a statement specifying that all of our examined specimens were adults.

line 208: to the base of the teeth ?

Agreed, we changed “bases” into “base”.

line 295: variants ?

Agreed, we changed “variant” into “variants”

Referee: 2

Comments to the Author(s):

I was pleased to re-review the manuscript, “Plicidentine and the repeated origins of snake venom fangs,” and am impressed at the work that has been done to address the major issues raised during the first round of review. I appreciate that the change in the FEA and the resultant change in the output has led to a shift in the discussion, but am worried about how the authors introduce their alternative hypothesis regarding tooth attachment. This seems like a likely scenario, but not

necessarily, as stated in the text, the only remaining possibility. This, coupled with some of the wording regarding the hypothesized ancestral condition does still feel like over-selling. That said, it's a great study and I'm excited to see it published.

Additional Comments:

Abstract:

Lines 28 & 33: "advanced" is a pretty loaded term, and doesn't really get at the point. Would it be better to say 'venomous'?

Agreed, we replaced "advanced" with "venomous".

Materials & methods

Line 82: It appears that the authors have replaced "μ" throughout the rest of the manuscript with "micro"

Thank you for pointing this out, we indeed decided to use the term microCT throughout the manuscript. For consistency we have now replaced "μCT" with "microCT" also on this line.

FEA section: Much improved overall! The only suggestion I have is that it might be informative here to provide information regarding the scan resolution, even if it can also be found in the supplemental info

Agreed, we have now added that information on lines 105-108: "The 3D geometries of the fang of *F. leucobalia* (SAMA R26990) for the biomechanical testing of the groove and plicidentine were extracted in Mimics v. 23.0 software (Materialise, Leuven, Belgium) using a combination of automatic (thresholding) and manual segmentation of CT data acquired at a resolution of 4.034 μm (electronic supplementary material, table S1)."

Line 126: "anteriorly on the maxilla" makes it sound like only the anterior-most portion of the maxilla is constrained. In looking at the supplemental figure it appears that all of the maxilla immediately underlying [overlying?] the fang is constrained instead. Is there a better way to describe this?

The 3D model of the maxilla was cut anteriorly, and the constraint was placed on the maxilla along that flat surface (i.e. "anteriorly on the maxilla"). There is no constraint immediately above or below the fang. To clarify this point we have added a line to our Materials and Methods (lines 108-110) stating that: "Only the portion of the maxilla immediately dorsal to the fang was retained, and the rest was digitally removed producing a vertical cutting plane." Additionally, on line 132 we now have: "Each fang was constrained (all translations and rotations) anteriorly on the maxilla (i.e. on the vertical cutting plane)..."

Lines 136-138: This bit may need a bit more explanation. What I'm reading indicates that the Von Mises stresses were only used to compare resistance to compression in grooved vs non-grooved fangs, and that the principal strain data were only used to compare bending in fangs with and without plicidentine?

We apologise for the confusion, we compared both Von Mises stresses and principal strains for all models. The word "respectively" on line 137 (now line 145) should not have been there, we have now removed it and the sentence reads as follows: "To evaluate the biomechanical importance of the groove and plicidentine (i.e. whether they increase resistance to compression and bending), we

compared Von Mises stress and principal strain regimes between the grooved and non-grooved models and models with and without plicidentine.”

Discussion

Lines 313-314: “These results leave only one possibility” is a very strong statement, doesn’t account for other possible alternative hypotheses, and implies a conclusion with no data to support it.

“These results eliminate all but one of the proposed hypotheses” gets the same information across and doesn’t overstate the findings of the current study.

Agreed, as suggested by the reviewer we have replaced the sentence “These results leave only one possibility among the functions proposed for plicidentine” with “These results eliminate all but one of the proposed hypotheses: plicidentine...” on what is now line 324.

Supplemental data

S Fig 1: lining up the loads with the different fang treatments is a bit misleading, is there a way to break these apart? Draw a line between the two? Put the loads along the bottom of the fig?

Agreed, we have added a vertical line to separate the different fang treatments from the loads.